# Restraint Stress and Repeated Corticosterone Administration Differentially Affect Neuronal Excitability, Synaptic Transmission and 5-HT_7_ Receptor Reactivity in the Dorsal Raphe Nucleus of Young Adult Male Rats

**DOI:** 10.3390/ijms232214303

**Published:** 2022-11-18

**Authors:** Joanna Bąk, Bartosz Bobula, Grzegorz Hess

**Affiliations:** Department of Physiology, Maj Institute of Pharmacology, Polish Academy of Sciences, Smetna 12, 31-343 Krakow, Poland

**Keywords:** 5-HT_7_ receptor, SB 269970, stress, corticosterone, dorsal raphe nucleus, glutamatergic transmission, GABAergic transmission, serotonin

## Abstract

Exogenous corticosterone administration reduces GABAergic transmission and impairs its 5-HT_7_ receptor-dependent modulation in the rat dorsal raphe nucleus (DRN), but it is largely unknown how neuronal functions of the DRN are affected by repeated physical and psychological stress. This study compared the effects of repeated restraint stress and corticosterone injections on DRN neuronal excitability, spontaneous synaptic transmission, and its 5-HT_7_ receptor-dependent modulation. Male Wistar rats received corticosterone injections for 7 or 14 days or were restrained for 10 min twice daily for 3 days. Repeated restraint stress and repeated corticosterone administration evoked similar changes in performance in the forced swim test. They increased the frequency of spontaneous excitatory postsynaptic currents (sEPSCs) recorded from DRN neurons. In contrast to the treatment with corticosterone, restraint stress-induced changes in sEPSC kinetics and decreased intrinsic excitability of DRN neurons did not modify inhibitory transmission. Repeated injections of the 5-HT_7_ receptor antagonist SB 269970 ameliorated the effects of restraint on excitability and sEPSC frequency but did not restore the altered kinetics of sEPSCs. Thus, repeated restraint stress and repeated corticosterone administration differ in consequences for the intrinsic excitability of DRN projection neurons and their excitatory and inhibitory synaptic inputs. Effects of repeated restraint stress on DRN neurons can be partially abrogated by blocking the 5-HT_7_ receptor.

## 1. Introduction

Chronic stress results in hyperactivation of the hypothalamic–pituitary–adrenal (HPA) axis, which contributes to the pathophysiology of mood disorders [1,2,3]. Acute and chronic stressors affect serotonin (5-HT) neurotransmission by modifying 5-HT release and reuptake, changing extracellular 5-HT levels, and altering the expression of 5-HT receptors in different brain structures [4,5,6]. These effects have been attributed to the activation of glucocorticoid receptors in the dorsal raphe nucleus (DRN) [7,8], the main source of widespread, long-range 5-HT projections to the forebrain. However, the influence of different forms of repeated stress on the functions of the DRN is not fully understood.

The activity of DRN 5-HT projection neurons remains under the control of a complex system of glutamatergic and GABAergic synaptic inputs. The excitatory afferents to the DRN originate in the prefrontal cortex and several subcortical sources [9,10]. Inhibitory synaptic inputs to DRN 5-HT projection neurons are provided mainly by local GABAergic interneurons [11,12], but the DRN also receives long-range GABAergic innervation from other brain structures [13]. Using an animal model of stress-induced depression-like behavior in rodents—repeated corticosterone administration [14]—we had previously investigated the influence of corticosterone on inhibitory synaptic transmission in the DRN. We found that repeated corticosterone administration for 7 and 14 days decreased the frequency of spontaneous inhibitory postsynaptic currents (sIPSCs) recorded from DRN projection neurons but did not modify the excitability of these neurons [15,16]. We also showed that corticosterone treatment lasting for 7 days increased the frequency of spontaneous excitatory postsynaptic currents (sEPSCs) in DRN projection neurons [16]. Corticosterone administration, however, mimics only one aspect of the organism’s stress response, namely the elevation of corticosterone plasma levels. It does not engage all components of the stress response [17]. Repeated restraint is an experimental paradigm used to evoke behavioral modifications in response to full activation of the stress response [18,19,20,21]. We have previously found enhanced excitatory synaptic activity recorded from neurons in hypothalamic and frontal cortical slices of rats that had been restrained for 10 min, twice daily for 3 days [22,23,24]. In the present study, we aimed to compare the effects of this pattern of repeated restraint on the glutamatergic input to DRN projection neurons and their intrinsic excitability with those of corticosterone treatment lasting 7 and 14 days.

Among several 5-HT receptor subtypes, the 5-HT_7_ receptor (5-HT7R) is particularly abundant in the DRN [25]. Activation of the 5-HT7R modifies the activity of the DRN neuronal network by increasing the excitability of local inhibitory interneurons and thus indirectly reducing the spiking activity of rat DRN projection cells [12]. We had previously shown that corticosterone treatment lasting 14 days dampened basal inhibitory transmission in the DRN and blunted the 5-HT7R-mediated increase in sIPSC frequency in projection neurons. Administration of the 5-HT7R receptor antagonists SB 269970, concurrent with corticosterone, prevented these effects [15]. Since 5-HT7Rs also appear to modulate sEPSCs in DRN projection neurons [16], the second aim of this study was to verify whether SB 269970 administration could ameliorate the repeated restraint-induced modifications of the excitatory synaptic input and its reaction to the activation of the 5-HT7R in DRN neurons.

## 2. Results

### 2.1. Comparison of the Effects of Repeated Corticosterone Administration and Repeated Restraint on the Forced Swim Test

The effects of repeated corticosterone administration and repeated restraint on performance on the forced swim test (FST) tested in Experiment 1 (see Section 4) were similar. Rats receiving corticosterone for 7 days (termed: 7d Cort) demonstrated shortened climbing time during the second trial of the FST in comparison to control animals receiving vehicle (termed: 7d Tw; 7d Tw vs. 7d Cort: 178.80 ± 7.02 s vs. 76.10 ± 5.07 s; t = 11.86, df = 18, *p* < 0.001; *t*-test; Figure 1a). Moreover, rats receiving repeated corticosterone injections for 7 days demonstrated shortened mobility time (7d Tw vs. 7d Cort: 184.00 ± 6.44 s vs. 80.90 ± 5.32 s; t = 12.35, df = 18, *p* < 0.001; *t*-test) and prolonged immobility time (7d Tw vs. 7d Cort: 116.00 ± 6.44 s vs. 219.10 ± 5.32 s; t = 12.35, df = 18, *p* < 0.001; *t*-test; Figure 1a).

Rats subjected to repeated corticosterone administration for 14 days (termed: 14d Cort) showed shortened climbing time in comparison to control animals receiving vehicle (termed: 14d Tw; 14d Tw vs. 14d Cort: 190.40 ± 9.42 s vs. 80.60 ± 6.88 s; t = 9.415, df = 18, *p* < 0.001; *t*-test; Figure 1b). These animals also exhibited shortened mobility time (14d Tw vs. 14d Cort: 198.00 ± 9.08 s vs. 84.10 ± 7.11 s; t = 9.879, df = 18; *t*-test) and prolonged immobility time (14d Tw vs. 14d Cort: 102.00 ± 9.08 s vs. 215.90 ± 7.11 s; t = 9.879, df = 18, *p* < 0.001; *t*-test; Figure 1b).

Similarly, animals subjected to repeated restraint twice daily for 3 days (termed: RS) demonstrated significantly shortened climbing time in comparison to control, handled animals (termed: hand) during the second trial of the FST (hand vs. RS: 180.70 ± 8.79 s vs. 110.20 ± 4.78 s; t = 7.05, df = 18, *p* < 0.001; *t*-test; Figure 1c). RS rats exhibited shortened mobility time (hand vs. RS: 187.40 ± 8.89 s vs. 113.40 ± 4.67 s; t = 7.37, df = 18, *p* < 0.001; *t*-test) and prolonged immobility time (hand vs. RS: 122.60 ± 9.30 s vs. 186.60 ± 4.67 s; t = 6.15, df = 18, *p* < 0.001; *t*-test; Figure 1c) as well.

### 2.2. Comparison of the Effects of Repeated Corticosterone Administration and Repeated Restraint on Intrinsic Excitability of DRN Neurons and Their Synaptic Inputs

In Experiment 1, all cells subjected to analysis expressed responses characteristic of putative DRN 5-HT projection neurons (Figure 2a_1_) [12]. The resting membrane potential, input resistance, and gain of DRN neurons originating from animals treated with corticosterone for 7 or 14 days did not differ from cells obtained from animals receiving vehicle (Table 1; Figure 2a_2_,b_1_,c_1_), [15]. In contrast, neurons in slices prepared from rats subjected to restraint expressed reduced gain in comparison to control, handled rats (t = 9.197, df = 45, *p* < 0.001; *t*-test; Figure 2d_1_). The resting membrane potential and input resistance of these cells were unchanged (Table 1).

Increased mean frequency of sEPSCs was evident after repeated administration of corticosterone for 7 and 14 days (7d Cort: t = 7.690, df = 38, *p* < 0.001; 14d Cort: t = 6.254, df = 40, *p* < 0.001; *t*-test; Figure 2b_2_,c_2_; Table 2) but neither the mean amplitude of sEPSCs nor parameters characterizing the kinetics of sEPSCs were altered in these groups of animals (Figure 2b_2_,c_2_; Table 2). In cells originating from rats subjected to restraint, the mean frequency of sEPSCs was higher in comparison to handled animals (t = 20.160, df = 45, *p* < 0.001; *t*-test; Figure 2d_2_; Table 2), but in contrast to neurons from rats receiving corticosterone, both the rise time and the decay time constant of sEPSCs were shortened (t = 4.230, df = 45, *p* < 0.001 and t = 2.292, df = 45, *p* = 0.027, respectively; *t*-test; Table 2).

In the same sample of DRN neurons, a decrease in the mean frequency of sIPSCs was observed after repeated administration of corticosterone for 7 and 14 days (7 days: t = 8.243, df = 38, *p* < 0.001; 14 days: t = 10.22, df = 40, *p* < 0.001; *t*-test; Figure 2b_3_,c_3_; Table 3) and neither the mean amplitude of sIPSCs nor the parameters characterizing the kinetics of sIPSCs were modified (Table 3). This is in line with a previous study [15]. In contrast, in cells originating from rats subjected to restraint (RS), the mean frequency, mean amplitude, rise time, and decay time constant of sIPSCs were unchanged (Figure 2d_3_; Table 3).

### 2.3. SB 269970 Attenuates the Effects of Repeated Restraint on the Excitability of DRN Neurons and Spontaneous EPSCs

We had previously shown that the 5-HT7R antagonist, SB 269970, ameliorated the effects of repeated corticosterone administration for 14 days on GABAergic transmission and 5-HT7R reactivity in the DRN [15]. In Experiment 2, we tested the possibility of preventing restraint stress-induced effects on the functions of DRN neurons by treatment with SB 269970. Restraint stress and SB 269970 significantly affected the excitability of DRN neurons (ANOVA: F_(3,87)_ = 12.63, *p* < 0.001). In neurons originating from rats subjected to restraint for 3 days and receiving vehicle injections (0.9% NaCl; termed: RS + NaCl), the gain was significantly lower in comparison to handled, vehicle-injected control (termed: hand + NaCl; hand + NaCl vs. RS + NaCl: q = 7.691, df = 87, *p* < 0.001; Tukey test; Table 4). Administration of SB 269970, along with restraint (termed: RS + SB), prevented the occurrence of restraint-related changes in the excitability of DRN neurons (hand + NaCl vs. RS + SB: q = 1.791, df = 87, *p* = 0.587; RS + NaCl vs. RS + SB: q = 5.900, df = 87, *p* < 0.001; Tukey test; Table 4). Neither the resting membrane potential nor the input resistance of DRN neurons changed significantly due to either separate or joint exposure to restraint and/or SB 269970 (*V_m_*: ANOVA: F_(3,87)_ = 2.168, *p* = 0.098; R_m_: ANOVA: F_(3,87)_ = 0.673, *p* = 0.571; Table 4).

Restraint stress and SB 269970 altered the mean frequency of sEPSCs (Figure 3a) recorded from DRN neurons (ANOVA: F_(3,87)_ = 140.20, *p* < 0.001). In neurons originating from rats subjected to restraint and receiving vehicle injections (termed: RS + NaCl), the mean sEPSC frequency was higher than in control preparations (hand + NaCl vs. RS + NaCl: q = 24.97, df = 87, *p* < 0.001; Tukey Test; Figure 3b_1_; Table 5). Cells from rats receiving SB 269970 along with restraint (termed: RS + SB) showed markedly lower sEPSC frequency in comparison to cells from restrained animals receiving vehicle (RS + NaCl vs. RS + SB: q = 20.12, df = 87, *p* < 0.001; Tukey test). However, in the RS + SB group, the mean frequency of sEPSCs was still slightly higher than in the control group receiving handling and vehicle injections (hand + NaCl vs. RS + SB: q = 4.852, df = 87, *p* = 0.005; Tukey test; Figure 3b_1_; Table 5). Administration of SB 269970 alone did not affect mean sEPSC frequency (hand + NaCl vs. hand + SB: q = 0.117, df = 87, *p* = 0.999; Tukey test; Figure 3b_1_; Table 5). No differences in the mean sEPSC amplitude between groups were observed (ANOVA: F_(3,87)_ = 2.03, *p* = 0.116; Tukey test; Figure 3b_2_; Table 5).

Repeated restraint and SB 269970 also altered the rise time (ANOVA: F_(3,87)_ = 8.815, *p* < 0.001) and the decay time constant of sEPSCs (ANOVA: F_(3,87)_ = 8.603, *p* < 0.001; Figure 3c_1_). Restraint stress decreased both the rise time (hand + NaCl vs. RS + NaCl: q = 5.174, df = 87, *p* = 0.002; Tukey test; Figure 3c_2_; Table 5) and the decay time constant (hand + NaCl vs. RS + NaCl: q = 5.941, df = 87, *p* < 0.001; Tukey test; Figure 3c_3_; Table 5). Administration of SB 269970 did not prevent these changes in rise time (RS + NaCl vs. RS + SB: q = 1.321, df = 87, *p* = 0.787; hand + NaCl vs. RS + SB: q = 3.853, df = 87, *p* = 0.038; Tukey test; Figure 3c_2_; Table 5) and the decay time constant (RS + NaCl vs. RS + SB: q = 2.095, df = 87, *p* = 0.453; hand + NaCl vs. RS + SB: q = 3.846, df = 87, *p* = 0.039; Tukey test; Figure 3c_3_; Table 5).

Restraint stress and SB 269970 altered neither the mean frequency, mean amplitude, rise time, nor decay time constant of sIPSCs recorded from DRN neurons (ANOVA; frequency: F_(3,73)_ = 0.382, *p* = 0.767; amplitude: F_(3,73)_ = 2.585, *p* = 0.060; rise time: F_(3,73)_ = 0.100, *p* = 0.960; decay time constant: F_(3,73)_ = 1.981, *p* = 0.124; Table 6).

### 2.4. SB 269970 Counteracts the Effects of Repeated Restraint on 5-HT_7_ Receptor-Mediated Changes in the Frequency of Spontaneous EPSCs

The frequency of sIPSCs recorded from rat DRN projection neurons is modulated by the 5-HT_7_R [12]. To test the effect of 5-HT_7_ receptor activation on sEPSCs in these neurons and to investigate the influence of restraint and SB 269970 on this effect, 5-CT, an agonist of the 5-HT_7_ receptor (see methods), was added to the ACSF after the initial recording of the basal synaptic activity described above.

Two-way ANOVA with repeated measures demonstrated significant effects of 5-CT (F_(1,87)_ = 571.50, *p* < 0.001) and experimental group (F_(3,87)_ = 147.90, *p* < 0.001), albeit with no interaction between these factors (F_(3,87)_ = 1.137, *p* = 0.339), on the mean frequency of sEPSCs recorded from cells obtained from animals subjected to restraint/handled and/or injected with SB 269970. Activation of the 5-HT_7_R resulted in a significant reduction of sEPSC frequency in all groups (hand + NaCl: t = 11.41, df = 87, *p* < 0.001; hand + SB: t = 10.58, df = 87, *p* < 0.001; RS + NaCl: t = 12.79, df = 87, *p* < 0.001; RS + SB: t = 13.06, df = 87, *p* < 0.001, Sidak test; Table 7, Figure 4a_1_). The data show that restraint stress attenuated the effect of 5-HT_7_R activation on sEPSCs frequency (Table 7, Figure 4a_2_), and administration of SB 269970, along with restraint, prevented the occurrence of this restraint-related modification in the reaction of sEPSCs to 5-HT_7_R activation (Table 7, Figure 4a_2_). Activation of the 5-HT_7_R induced no changes in the mean sEPSC amplitude in any of the groups (not shown).

Two-way ANOVA with repeated measures demonstrated a significant effect of 5-CT (F_(1,73)_ = 1501, *p* < 0.001) on the mean frequency of sIPSCs recorded from cells obtained from animals subjected to restraint and/or SB 269970 administration. No effect of the experimental group (F_(3,73)_ = 0.492, *p* = 0.689) or interaction between these factors (F_(3,73)_ = 1.166, *p* = 0.329) were observed. Activation of the 5-HT_7_R resulted in a significant increase in the frequency of sIPSCs in all groups (hand + NaCl: t = 19.37, df = 73, *p* < 0.001; hand + SB: t = 17.72, df = 73, *p* < 0.001; RS + NaCl: t = 20.22, df = 73, *p* < 0.001; RS + SB: t = 20.21, df = 73, *p* < 0.001, Sidak test; Table 8, Figure 4b_1_,b_2_). There were no differences between groups in the mean sIPSC amplitude before and after 5-HT_7_R activation (not shown).

## 3. Discussion

Repeated corticosterone administration is regarded as a preclinical rodent model of chronic stress [14]. Repeated daily corticosterone administration for up to 21 days in a high-dose regimen (40 mg/kg per day) induces depressive behavior in the FST [26]. A lower dose of corticosterone (20 mg/kg per day), like that used in the present study, has also been reported to increase immobility in the FST 20 days after administration [27], and the present data show that this effect could already be observed after 7 days of corticosterone injections. On the other hand, in contrast to repeated corticosterone injections, restraint stress repeated daily for 21 days had no significant effect on immobility in the FST [28]. This observation may be attributed to the known effect of a progressive reduction (habituation) of the hypothalamic–pituitary–adrenal (HPA) axis activation with repeated exposure to the same (homotypic) stressor [29,30,31]. It has been reported that a restraint stress procedure identical to the one used in our study (10 min twice daily) and repeated over 3, 7, and 14 days resulted in a progressive reduction of the magnitude of plasma corticosterone surge that occurs in response to acute restraint [19]. Therefore, we used a repeated restraint regime that provides the strongest HPA axis activation in increased plasma corticosterone levels. It should be noted that the present results were obtained in male rats. It has been demonstrated that there are sex differences in the effects of repeated restraint in the dorsal raphe nucleus, including, e.g., 5-HT_1A_ receptor G-protein coupling responses [32]. Importantly, young adult male rats, as used in the present study, express more pronounced repeated restraint-induced changes in the number of TPH-positive DRN neurons than old males [33].

Although marked changes in performance in the forced swim test were evident in Experiment 1, we observed no changes in the input resistance and intrinsic excitability of DRN projection cells after corticosterone treatment lasting 7 or 14 days, consistent with our previous studies [15,16]. It has recently been reported that repeated corticosterone administration for 21 days decreased the intrinsic excitability in a subpopulation of rat DRN projection cells [34]. While we have not differentiated recorded DRN neurons, and therefore any possible modification specific to certain cells might not have been visible among the whole population, it should be noted that in that study, a daily dose of 40 mg/kg of corticosterone was used (vs. 20 mg/kg/day in our study). It is conceivable that any plausible change in the intrinsic excitability might only develop with longer corticosterone treatments with a higher dose than the one used in our study. It should also be noted that Prouty et al. [34] reported a decrease in the input resistance accompanied by reduced membrane excitability of cells. Unless the input resistance is normalized by cell capacitance to obtain the specific membrane resistance values, this phenomenon might distort the assessment of intrinsic excitability [35].

To our knowledge, our findings are the first to characterize the effects of repeated restraint on the intrinsic excitability of DRN projection cells. The results of Experiment 1 show that restraint stress reduced the gain factor of DRN putative 5-HT neurons, indicating a decreased intrinsic excitability of these cells with no change in their input resistance, which coincided with changes in forced swim test performance. It has recently been shown that repeated restraint increases the density of 5-HT_1A_ receptors, which promotes a reduction in neuronal excitability in the DRN of male rats [36]. Decreased excitability has been reported following repeated restraint stress in rat paraventricular nucleus [22,23,35] and mouse arcuate nucleus [37] of the hypothalamus. The mechanism of this effect is likely to involve enhanced potassium currents, e.g., slowing a decay of rapidly activating, rapidly inactivating voltage-gated potassium currents [38]. Our results show that increased corticosterone level *per se* is unlikely to induce changes in active membrane properties of DRN neurons and suggest that other factors related to activation of the stress response of the organism, as in the case of restraint, are involved in the observed reduction of the intrinsic excitability.

Both types of treatment used in the present study resulted in a superficially similar increase in the mean frequency of sEPSCs recorded from DRN neurons. While the present data show an increase in sEPSC frequency after corticosterone treatment, consistent with our previous findings [16], a decrease or no change in the frequency of sEPSCs has been reported in two subpopulations of DRN neurons [34]. Again, differences in experimental conditions between the two studies are likely to account for this discrepancy. We have previously found that repeated restraint and corticosterone administration enhance transmission in local excitatory connections in the frontal cortex [24,39]. Likely, this effect may also occur in the cortical input to DRN neurons and perhaps also in other excitatory synaptic inputs to the DRN. It was found that repeated restraint stress does not alter the strength of glutamatergic synapses in the DRN [40]. Still, it should be noted that in that study, the restraint regime was employed for a longer time, and it is conceivable that the lack of effect is related to the adaptation to restraint. Interestingly, in parvocellular neurons of the rat hypothalamic paraventricular nucleus, restraint stress repeated for 3 days increased the mean frequency of sEPSCs and decreased the rise time and the decay time constant of sEPSCs [22]. These effects appear to be related to changes in the activation and deactivation kinetics of postsynaptic AMPA receptors [41]. In the present study, repeated restraint stress, but not corticosterone treatment, resulted in similar changes in the kinetics of sEPSCs. Of note, it has been reported that single swim stress also increases the frequency of sEPSCs in DRN serotonergic neurons, but in this case, it was accompanied by an increase in sEPSC decay time [42]. It has been shown that noradrenaline, acting through α_1_ adrenergic receptors, controls glutamatergic synaptic transmission and plasticity in the DRN, and this mechanism is impaired after chronic restraint stress [40]. It is tempting to speculate that while increased plasma corticosterone levels increase glutamate release from presynaptic terminals, only mechanisms are activated during a complete stress response, particularly the sympathetic-adreno-medullar (SAM) axis, induce modifications on the postsynaptic side.

The results of Experiment 2 indicate that 5-HT7R activation decreases the frequency of sEPSCs recorded from DRN neurons. Since most, if not all, glutamatergic inputs to DRN projection neurons originate in other brain structures [9,10], it is unlikely that HT7Rs could directly modulate excitatory synaptic activity recorded from DRN neurons in slice preparations where afferent connections are cut. Moreover, activation of 5-HT7Rs increases the spiking activity of glutamatergic neurons [43,44], which increases spontaneous synaptic activity recorded from target cells. We have previously shown that 5-HT7Rs located on DRN GABAergic interneurons can regulate inhibitory synaptic activity in the DRN [12], which is consistent with present findings. GABA may gate glutamate release in the DRN since many GABAergic terminals are organized in synaptic triads with a glutamatergic terminal and a common postsynaptic target [45]. This particular synaptic arrangement may explain the decrease in sEPSC frequency that occurs as a secondary effect to an increase in the inhibitory synaptic activity in the DRN, observed after 5-HT7R activation with a bath-applied agonist.

We have previously found that repeated corticosterone administration for 14 days decreases GABAergic transmission within the DRN and that functional desensitization of 5-HT7Rs located on DRN GABAergic interneurons is responsible for the lack of 5-HT7R-mediated increase in the frequency of sIPSCs in DRN projection neurons after corticosterone treatment [15]. Present data indicate that inhibitory synaptic transmission in the DRN and its 5-HT7R-dependent modulation remain unchanged after repeated restraint stress, demonstrating another difference in the outcome of repeated corticosterone administration and repeated restraint on the functions of DRN neurons. A blunted inhibitory effect of 5-HT7R activation on sEPSC frequency observed after restraint stress is likely to result from a distorted balance between enhanced excitatory and unchanged inhibitory synaptic activity in the DRN after repeated restraint. Amelioration of the effects of restraint stress on sEPSC frequency and neuronal excitability by the 5-HT7R antagonist SB 269970 complements our earlier work that showed a reversal of detrimental effects of repeated corticosterone administration on GABAergic transmission in the DRN by SB 269970 [15]. In both cases, treatment with SB 269970 tends to restore the excitation/inhibition balance in the DRN neuronal network. However, the mechanisms underlying these effects require further investigation.

## 4. Materials and Methods

### 4.1. Animals

Male Wistar rats (Charles River Laboratories, Sulzfeld, Germany) weighing approx. 150 g on arrival were housed in groups in standard laboratory cages and maintained on a 12-h light/dark schedule (lights on between 7 a.m. and 7 p.m.) with free access to standard RM3 food (Special Diet Services, Witham, UK) and tap water. The experiments were performed on animals at 7–10 weeks. The experimental procedures were approved by the Local Ethics Committee for Animal Experiments at the Institute of Pharmacology, Polish Academy of Sciences in Krakow. They were carried out in accordance with the European Community guidelines for the use of experimental animals and the national law.

### 4.2. Experiment 1—The Effects of Corticosterone Administration and Restraint Stress on Performance on the Forced Swim Test and Electrophysiology of DRN Neurons

#### 4.2.1. Treatment

The animals were assigned to six groups. The number of animals in each group was 20. Rats of the first experimental group received corticosterone injections subcutaneously (dose: 10 mg/kg, volume: 1 mL/kg; in 1% solution of vehicle (Tween 80 in water), [33] twice daily for 7 days. The second group received corticosterone injections for 14 days. Control rats received 1% Tween 80 injections for 7 or 14 days, respectively. Rats from the fifth group were restrained in metal tubes (diameter: 55 mm) for 10 min, twice daily (beginning at 8:00 and 17:00) for 3 days [22]. The control group for this treatment included animals that were handled for 10 min each day but otherwise treated similarly to restrained rats. Altogether, 120 rats were used in Experiment 1.

After the end of treatment 10 animals randomly selected from each group were individually subjected to the FST. Rats were forced to swim in a cylinder (40 cm high, 18 cm in diameter) filled up to a height of 35 cm with warm water (25 °C). The first (pre-test) trial lasted for 15 min, while the second trial lasted for 5 min. There was a 24-h interval between the first and the second trial. During the second trial, the duration of immobility, total mobility, and climbing were measured [46].

#### 4.2.2. Slice Preparation and Whole-Cell Recording

Brain slices containing the DRN were prepared from the remaining 10 animals of each group 24 h after the last substance administration or restraint to avoid acute effects of corticosterone and stress. Rats were anesthetized with isoflurane (Aerrane; Baxter Polska, Warsaw, Poland) and decapitated. Their brains were quickly removed and placed in an ice-cold artificial cerebrospinal fluid (ACSF) of the following composition (in mM): NaCl (130), KCl (5), CaCl_2_ (2.5), MgSO_4_ (1.3), KH_2_PO_4_ (1.25), NaHCO_3_ (1.25) and D-glucose (10), bubbled with a mixture of 95% O_2_ and 5% CO_2_. Midbrain slices (300 µm) were cut in the coronal plane using a vibrating microtome (VT1000; Leica Microsystems, Wetzlar, Germany) and incubated in ACSF at 30 ± 0.5 °C for at least 3 h. Individual slices were placed in the recording chamber and superfused at 2.5 mL/min with warm (32 ± 0.5 °C), modified ACSF containing (in mM): NaCl (132), KCl (2), CaCl_2_ (2.5), MgSO_4_ (1.3), KH_2_PO_4_ (1.25), NaHCO_3_ (26) and D-glucose (10), bubbled with 95% O_2_–5% CO_2_. Recording pipettes were pulled from borosilicate glass capillaries (Harvard Apparatus, Holliston, MA, USA) using the P97 puller (Sutter Instrument, Novato, CA, USA). The pipette solution contained (in mM): K-gluconate (130), NaCl (5), CaCl_2_ (0.3), MgCl_2_ (2), HEPES (10), Na_2_-ATP (5), Na-GTP (0.4) and EGTA (1; osmolarity: 290 mOsm, pH = 7.2). Pipettes had an open tip resistance of approx. 6 MΩ.

Whole-cell recordings were obtained from the dorsal part of the midline region of the DRN. Neurons were visualized using the Axioscope 2 upright microscope (Carl Zeiss, Oberkochen, Germany) with Nomarski optics, a 40x water immersion lens, and an infrared camera. Signals were recorded using the SEC 05LX amplifier (NPI, Tamm, Germany), filtered at 2 kHz, and digitized at 20 kHz using Digidata 1440A interface and Clampex 10 software (Molecular Devices, San Jose, CA, USA). The firing characteristics of recorded cells were assessed using rectangular current pulses (400 ms) in the current clamp mode. Putative 5-HT DRN projection neurons were identified based on their response characteristics [47]. The relationship between injected current intensity and the number of spikes was plotted for each cell. The gain was determined as the slope of the straight line fitted to experimental data.

To record sEPSCs, cells were voltage-clamped at −76 mV, and after 15 min of stabilization, synaptic events were recorded for 4 min as inward currents. Next, cells were voltage-clamped at 0 mV, and after 15 min of stabilization, sIPSCs were recorded for 4 min as outward currents [12]. Spontaneous EPSCs and IPSCs were detected offline using the Mini Analysis Program v.6 (Synaptosoft, Decatur, GA, USA), and individual events were selected manually for further analysis. Data were accepted for the analysis when access resistance ranged between 15 and 18 MΩ and was stable (<25% change) during recordings. The threshold amplitude for the detection of sEPSCs was set to 6 pA and for sIPSCs to 10 pA. EPSC and IPSC kinetics were determined from averaged events for each cell. Rise time was measured as the time needed for the current to rise from 10 to 90% of the peak. The decay time constant (tau) was determined by fitting an exponential function to the decay phase of the current.

### 4.3. Experiment 2—Attenuation of the Effects of Repeated Restraint on the Excitability of DRN Neurons and Spontaneous EPSCs by a 5HT7R Antagonist

#### 4.3.1. Treatment

The animals were assigned to 4 groups. Rats of the first group were restrained in metal tubes (diameter: 55 mm) for 10 min, twice daily (beginning at 8 a.m. and 7 p.m.) for 3 days [22]. These animals received intraperitoneal injections of SB 269970 (dose: 2.5 mg/kg, volume: 1 mL/kg) once daily. Rats of the second group were restrained (10 min, twice daily for 3 days) and received intraperitoneal injections of 0.9% NaCl once daily. Control groups for this treatment included animals that were not restrained but instead were handled for 10 min, twice daily for 3 days and received either SB 269970 solution or 0.9% NaCl. The number of animals in each group was 10. Altogether, 40 rats were used in Experiment 2.

#### 4.3.2. Slice Preparation and Whole-Cell Recording

Brain slices were prepared, and cells were recorded as in Experiment 1. 5-carboxyamidotryptamine (5-CT, Tocris; 250 nM), a nonselective agonist of 5-HT_1A_ and 5-HT_7_ receptors was used to activate 5-HT7Rs. In order to block its effects on 5-HT_1A_ receptors, a selective 5-HT_1A_ antagonist N-[2-[4-(2-Methoxyphenyl)-1-piperazinyl]ethyl]-N-2-pyridinylcyclohexanecarboxamide (WAY 100635, Tocris, 2 µM) was added to ACSF. SB 269970 hydrochloride was obtained from Hello Bio.

### 4.4. Statistical Analysis

All statistical analyses were performed using GraphPad Prism 9 software (Graphpad Software Inc., San Diego, CA, USA). The analysis of behavioral and electrophysiological data in Experiment 1 was carried out using the Student’s *t*-test for independent samples. In Experiment 2, the analysis of basal parameters characterizing cell membrane properties and postsynaptic currents, after checking normality with the Shapiro-Wilk test, was carried out using one-way ANOVA followed by Tukey’s multiple comparison test. Data from recordings involving 5-HT_7_ receptor activation were analyzed using two-way ANOVA, followed by Sidak’s post hoc test. The percentage change relative to baseline was used as the dependent variable for estimating the effects on synaptic activity in slices from different treatment groups. The significance level was set at *p* = 0.05 for all comparisons. The results are expressed as the mean ± SEM.

## 5. Conclusions

In summary, this study demonstrates that while repeated restraint stress and repeated corticosterone administration evoke similar changes in performance on the forced swim test, these treatments differ in their consequences for the intrinsic excitability of DRN projection neurons and the excitatory and inhibitory synaptic inputs they receive. These findings provide a rationale for investigating the roles of the HPA and SAM axes in mediating the effects of chronic stress on the neuronal functions of the DRN.

## Figures and Tables

**Figure 1 ijms-23-14303-f001:**
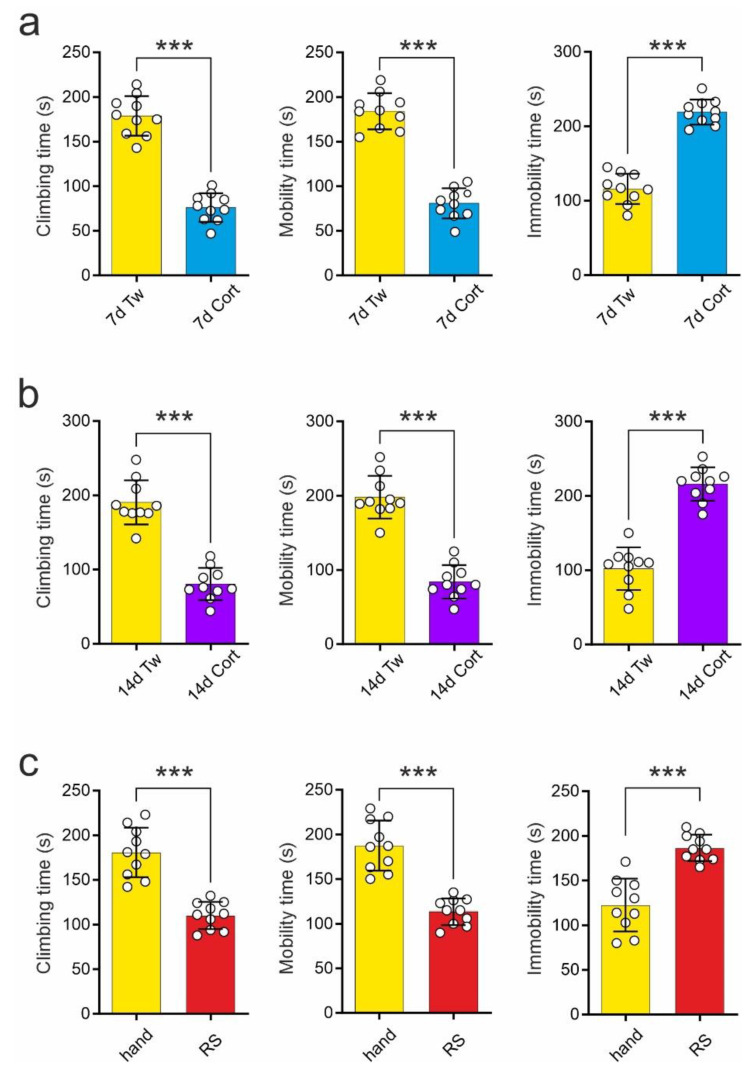
Repeated corticosterone administration lasting 7 days (7d Cort; (**a**)) or 14 days (14d Cort; (**b**)) and restraint stress (RS; (**c**)) result in a decrease in climbing time (**left**) and mobility time (middle) and in an increase in immobility time (**right**) in the forced swim test compared to control animals receiving vehicle (7d Tw or 14d Tw) or handling (hand), respectively. Shown are mean values ± SEM with circles representing each rat’s performance. *** *p* < 0.001.

**Figure 2 ijms-23-14303-f002:**
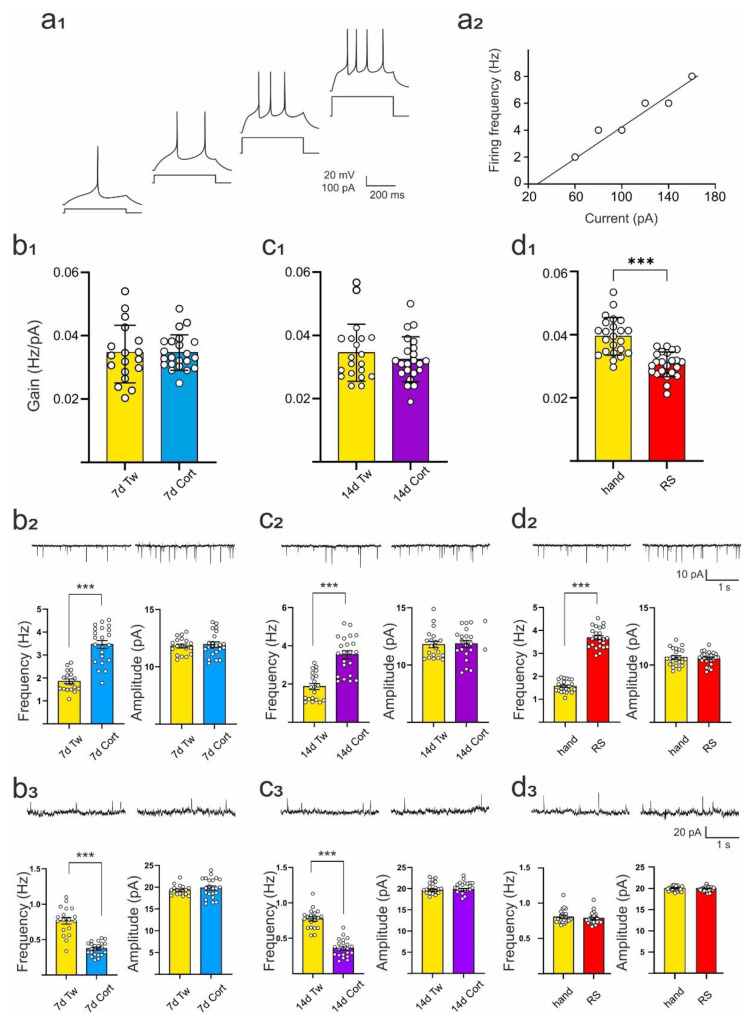
The effects of repeated corticosterone administration lasting 7 or 14 days and restraint stress on the excitability and spontaneous postsynaptic currents recorded from DRN neurons. (**a_1_**) Response of a representative putative 5-HT cell to depolarizing current pulses of increasing intensity; (**a_2_**) Relationship between spiking rate and injected current for the cell shown in (**a_1_**). The slope of the straight line fitted to experimental data represents gain; (**b_1_**,**c_1_**,**d_1_**) Graphs showing the effects of repeated corticosterone administration lasting 7 days (7d Cort) or 14 days (14d Cort) and restraint stress (RS) on the gain of DRN neurons; (**b_2_**,**c_2_**,**d_2_**) The effects of repeated corticosterone administration lasting 7 days or 14 days and restraint stress on the mean frequency (**left**) and mean amplitude (**right**) of sESPCs. Shown in the upper part of each panel are sample sEPSC recordings from representative neurons from each group; (**b_3_**,**c_3_**,**d_3_**) The effects of repeated corticosterone administration lasting 7 days or 14 days and restraint stress on the mean frequency (**left**) and mean amplitude (**right**) of sISPCs. Shown in the upper part of each panel shown are sample sIPSC recordings from representative neurons of each group. Bars represent mean values ± SEM, with circles representing individual cells. 7d Tw, 14d Tw—vehicle administered for 7 and 14 days, respectively; a hand—handling. *** *p* < 0.001.

**Figure 3 ijms-23-14303-f003:**
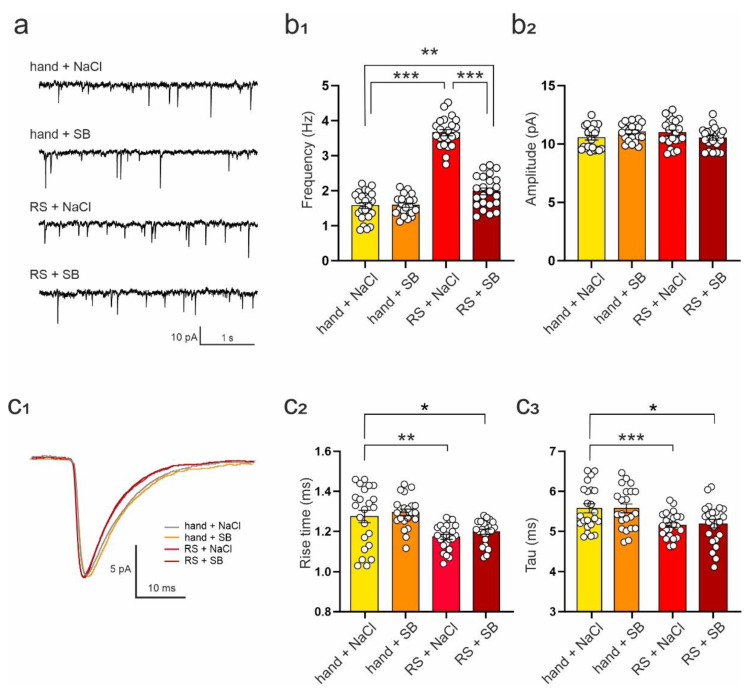
The influence of SB 269970 administration on restraint stress-induced effects on sEPSCs in DRN neurons. (**a**) Sample sEPSC recordings from representative neurons of each group; (**b_1_**) Mean (±SEM) sEPSC frequency. hand + NaCl vs. RS + NaCl: *** *p* < 0.001; RS + NaCl vs. RS + SB: *** *p* < 0.001; hand + NaCl vs. RS + SB: ** *p* < 0.01; (**b_2_**) Mean (± SEM) sEPSC amplitude; (**c_1_**) Superposition of averaged sEPSCs from representative neurons of each group; (**c_2_**) sEPSC rise time (mean ± SEM). hand + NaCl vs. RS + NaCl: ** *p* < 0.01; hand + NaCl vs. RS + SB: * *p* < 0.05. (**c_3_**) sEPSC decay time constant (mean ± SEM). Hand + NaCl vs. RS + NaCl: *** *p* < 0.001; hand + NaCl vs. RS + SB: * *p* < 0.05. In each bar, graph circles represent individual cells.

**Figure 4 ijms-23-14303-f004:**
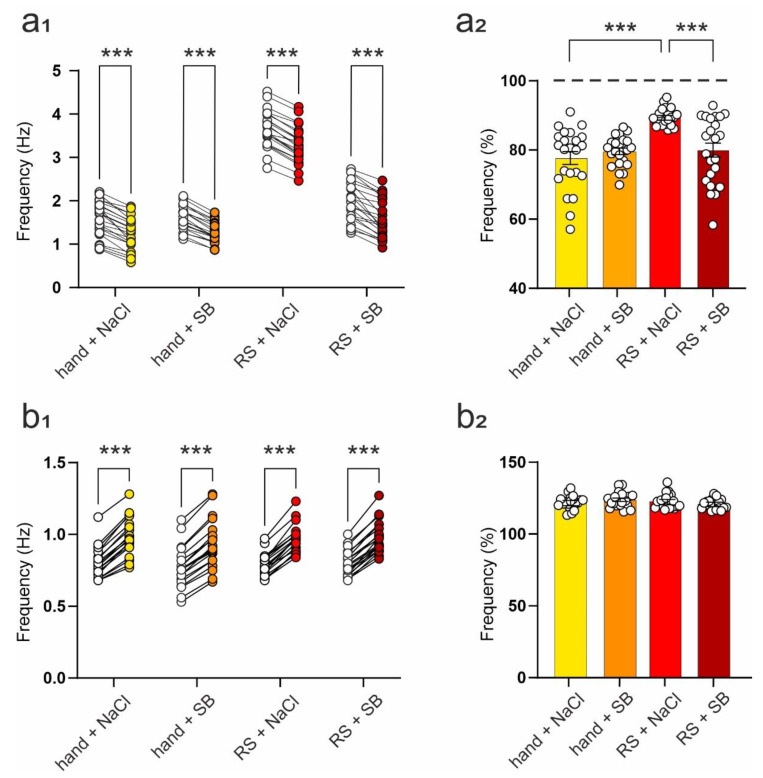
The influence of SB 269970 administration on restraint stress-induced changes in sEPSC (**a_1_**,**a_2_**) and sIPSC frequency (**b_1_**,**b_2_**) and the effects of 5-HT7R activation. (**a_1_**) The effects of 5-HT7R activation on the frequency of sEPSCs recorded from neurons originating from each group. For each cell, shown are mean sEPSC frequency values before (open circles) and after (filled circles) addition of 5-CT to the ACSF. *** *p* < 0.001; (**a_2_**) The effect of 5-HT7R activation on the sEPSC frequency (mean ± SEM) shown as a percentage of basal sEPSC frequency (before 5-CT addition). Hand + NaCl vs. RS + NaCl: *** *p* < 0.001; RS + NaCl vs. RS + SB: *** *p* < 0.001; (**b_1_**) The effects of 5-HT7R activation on the frequency of sIPSCs recorded from neurons originating from each group. For each cell, shown are mean sIPSC frequency values before (open circles) and after (filled circles) addition of 5-CT to the ACSF. *** *p* < 0.001; (**b_2_**) The effect of 5-HT7R activation on the sIPSC frequency (mean ± SEM) shown as a percentage of basal sEPSC frequency; *p* > 0.05. In each graph, circles represent individual cells.

**Table 1 ijms-23-14303-t001:** Effects of corticosterone and restraint stress on basic electrophysiological characteristics of recorded neurons (mean ± SEM).

Group	*V_m_* (mV)	*R_m_* (MΩ)	Gain (Hz/pA)	*n*
7d Tw	−68.78 ± 0.87	597.80 ± 22.33	0.0347 ± 0.0018	18
7d Cort	−69.36 ± 0.59	598.10 ± 16.76	0.0347 ± 0.0012	22
14d Tw	−67.10 ± 1.09	581.00 ± 15.79	0.0345 ± 0.0020	20
14d Cort	−67.36 ± 0.95	583.00 ± 13.24	0.0325 ± 0.0015	22
hand	−65.91 ± 0.69	558.20 ± 11.30	0.0387 ± 0.0008	23
RS	−65.82 ± 0.64	545.50 ± 11.57	0.0293 ± 0.0007 ***	24

*V_m_*—resting membrane potential; *R_m_*—input resistance; *n*—number of cells; *** *p* < 0.001, *t*-test.

**Table 2 ijms-23-14303-t002:** Effects of corticosterone and restraint stress on parameters characterizing sEPSCs (mean ± SEM).

Group	Mean Frequency (Hz)	Mean Amplitude (pA)	Rise Time (ms)	Decay Time Constant (*τ*, ms)	*n*
7d Tw	1.85 ± 0.10	11.82 ± 0.17	1.057 ± 0.022	4.912 ± 0.184	18
7d Cort	3.46 ± 0.17 ***	11.95 ± 0.22	1.046 ± 0.027	4.931 ± 0.157	22
14d Tw	1.87 ± 0.16	11.79 ± 0.27	1.053 ± 0.023	4.979 ± 0.208	20
14d Cort	3.53 ± 0.21 ***	11.85 ± 0.29	1.056 ± 0.036	4.962 ± 0.170	22
hand	1.56 ± 0.05	10.67 ± 0.15	1.221 ± 0.013	5.283 ± 0.093	23
RS	3.67 ± 0.09 ***	10.59 ± 0.12	1.146 ± 0.012 ***	5.050 ± 0.045 *	24

*n*—number of cells; * *p* < 0.05, *** *p* < 0.001, *t*-test.

**Table 3 ijms-23-14303-t003:** Effects of corticosterone and restraint stress on parameters characterizing sIPSCs (mean ± SEM).

Group	Mean Frequency (Hz)	Mean Amplitude (pA)	Rise Time (ms)	Decay Time Constant (*τ*, ms)	*n*
7d Tw	0.77 ± 0.05	19.33 ± 0.26	1.894 ± 0.092	6.967 ± 0.171	18
7d Cort	0.37 ± 0.02 ***	19.80 ± 0.48	1.873 ± 0.079	7.007 ± 0.159	22
14d Tw	0.77 ± 0.03	19.61 ± 0.30	1.878 ± 0.078	6.993 ± 0.183	20
14d Cort	0.36 ± 0.03 ***	19.77 ± 0.27	1.810 ± 0.051	6.993 ± 0.174	22
hand	0.80 ± 0.02	19.82 ± 0.10	1.899 ± 0.018	7.738 ± 0.084	23
RS	0.78 ± 0.02	19.71 ± 0.09	1.924 ± 0.023	7.733 ± 0.086	24

*n*—number of cells; *** *p* < 0.001, *t*-test.

**Table 4 ijms-23-14303-t004:** Effects of restraint and SB 269970 on basic electrophysiological characteristics of DRN neurons (mean ± SEM).

Group	*V_m_* (mV)	*R_m_* (MΩ)	Gain (Hz/pA)	*n*
hand + NaCl	−68.78 ± 0.87	597.80 ± 22.33	0.0347 ± 0.0018	23
hand + SB	−65.05 ± 0.62	554.80 ± 20.92	0.0394 ± 0.0014	22
RS + NaCl	−67.48 ± 0.96	545.30 ± 14.15	0.0309 ± 0.0008 ***	23
RS + SB	−67.35 ± 0.87	538.70 ± 16.80	0.0378 ± 0.0011 ***	23

*n*—number of cells; One–Way ANOVA with post hoc Tukey test; hand + NaCl vs. RS + NaCl: *** *p* < 0.001; RS + NaCl vs. RS + SB: *** *p* < 0.001.

**Table 5 ijms-23-14303-t005:** Effects of restraint and SB 269970 on parameters characterizing sEPSCs (mean ± SEM).

Group	Mean Frequency (Hz)	Mean Amplitude (pA)	Rise Time (ms)	Decay Time Constant (*τ*, ms)	*n*
hand + NaCl	1.58 ± 0.08 **	10.54 ± 0.20	1.274 ± 0.031 *	5.576 ± 0.114 *	23
hand + SB	1.59 ± 0.06	11.05 ± 0.17	1.295 ± 0.017	5.575 ± 0.101	22
RS + NaCl	3.66 ± 0.09 ***	10.98 ± 0.23	1.172 ± 0.013 **	4.980 ± 0.066 ***	23
RS + SB	1.98 ± 0.09 ***	10.52 ± 0.18	1.198 ±0.013	5.190 ± 0.107	23

*n*—number of cells; One–Way ANOVA with post hoc Tukey test; hand + NaCl vs. RS + NaCl: ** *p* < 0.01, *** *p* < 0.001; RS + NaCl vs RS + SB: *** *p* < 0.001; hand + NaCl vs. RS + SB: * *p* < 0.05, ** *p* < 0.01.

**Table 6 ijms-23-14303-t006:** Effects of restraint and SB 269970 on parameters characterizing sIPSCs (mean ± SEM).

Group	Mean Frequency (Hz)	Mean Amplitude (pA)	Rise Time (ms)	Decay Time Constant (*τ*, ms)	*n*
hand + NaCl	0.80 ± 0.02	19.61 ± 0.17	1.908 ± 0.050	7.783 ± 0.166	19
hand + SB	0.77 ± 0.03	19.30 ± 0.24	1.944 ± 0.074	7.500 ± 0.126	19
RS + NaCl	0.79 ± 0.02	19.92 ± 0.14	1.944 ± 0.040	7.895 ± 0.158	19
RS + SB	0.80 ± 0.02	19.85 ± 0.13	1.933 ± 0.044	7.963 ± 0.128	20

*n*—number of cells; One–Way ANOVA: *p* > 0.05.

**Table 7 ijms-23-14303-t007:** Effects of 5-HT_7_ receptor activation on parameters characterizing sEPSCs in DRN neurons from rats receiving restraint and SB 269970 (mean ± SEM).

Group	Mean Frequency (Hz)			*n*
	Before 5-CT	With 5-CT	% of Baseline	
hand + NaCl	1.58 ± 0.08	1.24 ± 0.08 ***	77.68 ± 1.85	23
hand + SB	1.59 ± 0.06	1.26 ± 0.05 ***	79.62 ± 0.94	22
RS + NaCl	3.66 ± 0.09	3.28 ± 0.09 ***	89.44 ± 0.51 ***	23
RS + SB	1.98 ± 0.09	1.59 ± 0.09 ***	79.98 ± 2.02 ***	23

*n*—number of cells; RM Two–Way ANOVA with post hoc Sidak test: *** *p* < 0.001. For percentage values: One–Way ANOVA with post hoc Tukey test: hand + NaCl vs. RS + NaCl: *** *p* < 0.001; RS + NaCl vs. RS + SB: *** *p* < 0.001.

**Table 8 ijms-23-14303-t008:** Effects of 5-HT_7_ receptor activation on parameters characterizing sIPSCs in DRN neurons from rats receiving restraint and SB 269970 (mean ± SEM).

Group	Mean Frequency (Hz)			*n*
	Before 5-CT	With 5-CT	% of Baseline	
hand + NaCl	0.80 ± 0.02	0.98 ± 0.03 ***	122.20 ± 1.07	19
hand + SB	0.77 ± 0.03	0.93 ± 0.04 ***	121.40 ± 0.83	19
RS + NaCl	0.79 ± 0.02	0.97 ± 0.02 ***	123.90 ± 1.22	19
RS + SB	0.80 ± 0.02	0.98 ± 0.02 ***	122.90 ± 1.11	20

*n*–number of cells; RM Two–Way ANOVA with post hoc Sidak test: *** *p* < 0.001. For percentage values: One–Way ANOVA: *p* > 0.05.

## Data Availability

The data presented in this study are available on request from the corresponding author.

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
