# Peer review of "Restraint Stress and Repeated Corticosterone Administration Differentially Affect Neuronal Excitability, Synaptic Transmission and 5-HT7 Receptor Reactivity in the Dorsal Raphe Nucleus of Young Adult Male Rats"

_ijms, 2022, doi:10.3390/ijms232214303_

Round 1

Reviewer 1 Report

In their study, Joanna Bak and colleagues demonstrate different effects of repeated corticosterone administration and repeated restraint stress on the function of DRN neurons. The subject matter is interesting. The manuscript is well-written and worth publishing. I really enjoyed reading. The experiments made and the statistical analyses are comprehensible. I have only a few minor comments that would increase the accessibility of the manuscript.

Figures: 

(1) Figure 1 to 4: Many journals do not allow red and green to be combined in figures because a significant portion of people is red-green blind. Perhaps the authors want to take that into account here as well?

(2) The labels are between figures very different in size! This is not very appealing.

(3) Some labels are so small that I can barely read them (e.g., Figure 2d2, 3a, scale labeling)

(4) The line thickness is partly too thin (e.g. Figure 2d3, 3c2, brackets)

Tables: Why are you using different labels (e.g., ### or ***) to say the same? Why not uniform?

Line 19: In what was the frequency increased? DRN neurons? Please specify!

Line 404: How many animals were used in total? Please name the number?

Line 407: (lights on between 7am and 7pm)

Line 408: Standard food? Please name the manufacturer and the name of the product!

Line 414, 472: A headline without content. Wouldn't it be possible to find a title that already provides the reader with information?

Line 422, 476: (beginning at 8am and 5pm)

Line 430: Delete the second “and climbing”

Line 494: Do you really used GraphPad Prism 7? For me graphs look very much like from GraphPad Prism 9.

Reviewer 2 Report

COMMENTS TO THE EDITORS AND THE AUTHORS

Manuscript ID ijms-021357: “Restraint stress and repeated corticosterone administration differentially affect neuronal excitability, synaptic transmission and 5-HT7 receptor reactivity in the rat dorsal raphe nucleus"

Please find below some of the comments for the above-mentioned manuscript.

SUMMARY OF THE CONTENT

The authors stated that the study aimed to compare the effects of repeated restraint stress and repeated corticosterone injections on dorsal raphe nucleus (DRN) neuronal excitability, spontaneous synaptic transmission and its 5-HT7 receptor-dependent modulation. The corticosterone injections were given to male Wistar rats (7 or 14 days) or male were restrained for 10 min twice daily for 3 days. The results showed that the repeated restraint stress and the repeated corticosterone administration evoked similar changes in performance in the forced swim test and increased the frequency of spontaneous excitatory postsynaptic currents (sEPSCs). The restraint stress induced changes in sEPSC kinetics and decreased intrinsic excitability of DRN neurons but did not modify inhibitory transmission. The repeated injections of the 5-HT7 receptor antagonist (SB 269970) ameliorated the effects of restraint on excitability and sEPSC frequency but did not restore the altered kinetics of sEPSCs.

THE OVERALL OPINION OF THE MANUSCRIPT

The strengths: the manuscript presents new and interesting knowledge; the results were obtained from the in vivo experiments; the figures very clearly present the results.

The limitations: the title is not precisely formulated; the citation of the original, and important pioneered results, as well as recent advance in the field focusing on the subject of the study are missing; the age of animals is not provided; the real controls (without any treatment with vehicle/NaCl) in experiments with treatments are missing; the intra- and inter- assay coefficients are not provided.

 Please find enclosed some of the suggestions in the comments to the authors listed below.

(1) TITLE

Please consider modifying the title since the results are obtained on the males of particular age. In the present form (“…in the rat dorsal raphe nucleus”) the title has general meaning.

(2) INTRODUCTION

2.1. Please describe the original, and important pioneered results, as well as recent advance in the field focusing on the subject of the study. 

(3) MATERIALS AND METHODS

3.1. Please provide the Key resources table.

3.2. Please provide the age of the animals.

3.3. Please describe how do you calculated the number of animals to achieve accurate statistical analyses.

3.4. Please perform the experiments using the real controls (without any treatment with vehicle/NaCl) in experiments with treatments.

3.5. Please provide intra- as well as inter-assay coefficients for all analyses.

(4) RESULTS

4.1. Please provide results and figures for the new measurements.

(5) DISCUSSION

5.1. Please discuss the original, and important pioneered results, as well as recent advance in the field focusing on the subject of the study. 

5.2. Please discuss the new results.

5.3. Please discuss the limitation of the study related to the sex and age.

(6) REFERENCES

6.1. Please provide references describing the original, and important and pioneered results, but also references describing the recent advance in the field. 

(7) FIGURES and FIGURE LEGENDS

7.1. Please provide new figures and figure legends showing the new results.

I would greatly appreciate if you will contact me if you find something in my comments is missing/unclear/incorrect.

Good luck and all the best J

Round 2

Reviewer 2 Report

The authors improved the manuscript.